# Striatal dynamics explain duration judgments

**Thiago S Gouvêa[†], Tiago Monteiro[†], Asma Motiwala, Sofia Soares, Christian Machens, Joseph J Paton***

Champalimaud Neuroscience Programme, Champalimaud Centre for the Unknown, Lisbon, Portugal

**Abstract** The striatum is an input structure of the basal ganglia implicated in several time-dependent functions including reinforcement learning, decision making, and interval timing. To determine whether striatal ensembles drive subjects' judgments of duration, we manipulated and recorded from striatal neurons in rats performing a duration categorization psychophysical task. We found that the dynamics of striatal neurons predicted duration judgments, and that simultaneously recorded ensembles could judge duration as well as the animal. Furthermore, striatal neurons were necessary for duration judgments, as muscimol infusions produced a specific impairment in animals' duration sensitivity. Lastly, we show that time as encoded by striatal populations ran faster or slower when rats judged a duration as longer or shorter, respectively. These results demonstrate that the speed with which striatal population state changes supports the fundamental ability of animals to judge the passage of time.

**\*For correspondence:** joe.paton@
neuro.fchampalimaud.org

[†]These authors contributed
equally to this work

**Competing interests:** The
authors declare that no
competing interests exist.

**Reviewing editor:** Timothy
Behrens, Oxford University,
United Kingdom

## Introduction

Time, like space, is a fundamental dimension of the environment, yet how it is processed in the brain is poorly understood. A number of recent studies have identified dynamics that allow for robust representation of time by populations of neurons in multiple areas including the hippocampus (*Pastalkova et al., 2008*; *MacDonald et al., 2011*), prefrontal (*Kim et al., 2013*; *Xu et al., 2014*), parietal (*Leon and Shadlen, 2003*; *Janssen and Shadlen, 2005*) and motor (*Lebedev et al., 2008*) cortices, cerebellum (*Mauk and Buonomano, 2004*), and the striatum (*Matell et al., 2003*; *Jin et al., 2009*; *Adler, 2012*; *Mello et al., 2015*). However, any dynamics that result in a continuously-evolving and non-repeating population state can be used to encode time (*Buonomano, 2014*), and it is not known whether such temporal representations would inform subjects' judgments of duration or merely covary with elapsing time. The striatum, a brain structure known to be involved in reinforcement learning and decision making (*Lau and Glimcher, 2008*; *Samejima et al., 2005*; *Lee et al., 2015*), has been implicated in interval timing by several lines of evidence (*Hinton and Meck, 2004*; *Harrington et al., 2009*; *Wencil et al., 2010*; *Malapani, 1998*; *Meck, 2006*). However, whether dynamics in striatal activity can explain the perceptual performance of behaving subjects is unknown. To determine whether striatal ensembles drive subjects' judgments of duration, we manipulated and recorded from striatal neurons in rats performing a duration categorization psychophysical task.

## Results

To measure the duration sensitivity of subjects' timing judgments, we trained rats to judge whether time intervals belonged to a long or short category (*Gouvêa et al., 2014*) (see Materials and methods; *Figure 1a*). At each self-initiated trial, two brief tones (interval onset,

**eLife digest** You know someone is a good cook from their rice - grains must be well cooked, but not to the point of being mushy. Despite consistently using the same pot and stove, we, however, will sometimes overcook it. It is as if our inner sense of time itself is variable. What is it about the brain that explains this variability in time estimation and indeed our ability to estimate time in the first place?

One issue the brain must confront in order to estimate time is that individual brain cells typically fire in bursts that last for tens of milliseconds. So how does the brain use this short-lived activity to track minutes and hours? One possibility is that individual neurons in a given brain region are programmed to fire at different points in time. The overall firing pattern of a group of neurons will therefore change in a predictable way as time passes.

Gouvêa, Monteiro et al. found such predictably changing patterns of activity in the striatum of rats trained to estimate and categorize the duration of time intervals as longer or shorter than 1.5 seconds. Interestingly, when rats mistakenly categorized a short interval as a long one, population activity had travelled farther down its path than it would normally (and vice-versa for long intervals incorrectly categorized as short), suggesting that variability in subjective estimates of the passage of time might arise from variability in the speed of a changing pattern of activity across groups of neurons.

As further evidence for the involvement of the striatum, inactivating the structure impaired the rats' ability to correctly classify even the longest and shortest interval durations.

The next challenge is to determine exactly how the striatum generates these time-keeping signals, at which stage variability originates, and how the brain regions that the striatum signals to use them to control an animal's behavior.

offset) were presented separated in time by an interval randomly selected from the set I = {0.6, 1.05, 1.26, 1.38, 1.62, 1.74, 1.95, 2.4} seconds. Judgments about interval duration were reported at two laterally located nose ports: choosing the left side was reinforced with water after intervals longer than 1.5 seconds (long stimuli), and the right side otherwise (short stimuli, *Figure 1b*). Animals were required to withhold choice until interval offset. Animals made virtually no errors when categorizing the easiest (i.e. shortest and longest) intervals, but categorization performance declined as intervals approached the 1.5 second categorical boundary (*Figure 1c*).

We recorded action potentials during task performance (see Materials and methods), from populations of single striatal neurons targeting dorsal-central striatum, an area where manipulations produced timing deficits (*Meck, 2006*) (*Figure 2a*. For a reconstruction of striatal recording sites see *Figure 2—figure supplement 1*.). We observed that striatal neurons displayed diverse firing patterns, with different units firing at different times within the interval period (*Figure 2b–d*). Can such firing patterns support duration judgments? To determine whether and the degree to which individual neurons could contribute to duration judgments, for each trial, we counted spikes in the last 500 ms of the interval period and compared spike count distributions of short vs long stimulus trials using a receiver operating characteristic (ROC) analysis (see Materials and methods). We found that the majority of neurons (~57%) preferred either short or long stimuli (*Figure 2e*; short-preferring: n = 159/433, 36.7%; long-preferring: n = 87/433, 20.1%; permutation test, p<0.05). As expected, short-preferring neurons displayed higher firing on average prior to the 1.5 s category boundary, after which long-preferring neurons displayed higher firing (*Figure 2f*). These averaged activity patterns resemble the likelihood of receiving reward on a moment-by-moment by basis should the animal choose short or long (compare with reward contingency in *Figure 1b*). Such signals, previously observed in the parietal cortex of monkeys performing a similar timing task (*Leon and Shadlen, 2003*) and in the striatum in a value based decision task (*Lau and Glimcher, 2008*), are potentially useful for guiding choice. However, were animals' judgments indeed guided by such signals, it should be possible to predict choices reported later in the trial using neural activity collected during interval stimuli. Indeed, in trials wherein a near boundary interval was judged as long, firing of the short (long) preferring subpopulation dropped (rose) faster, so that the two curves crossed before the 1.5 s boundary (*Figure 2g*). Conversely, in trials wherein the same interval was judged as short,

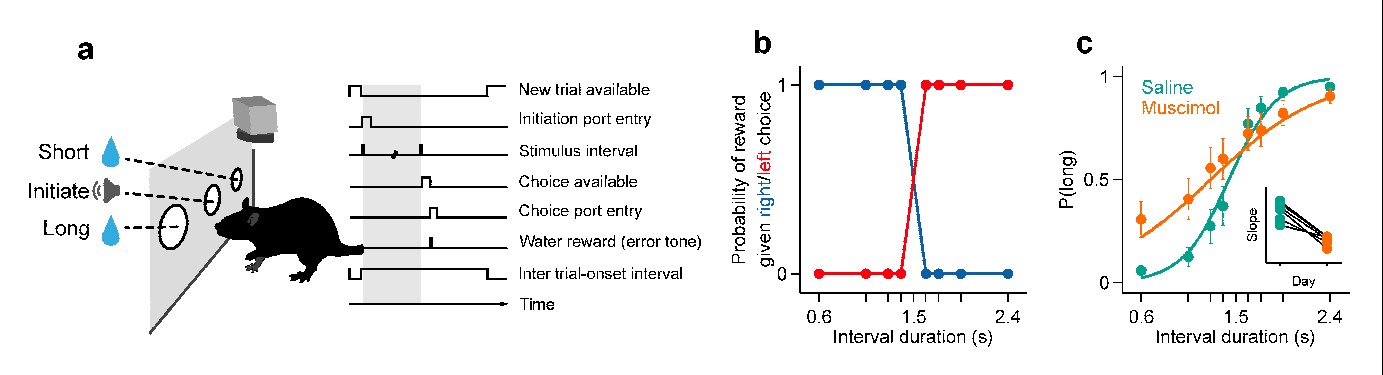

**Figure 1.** Rats judged interval durations as either long or short. (a) Rats triggered interval stimuli (i.e. two brief auditory tones separated by a silent interval of random duration) by inserting their snout into a central port. Following interval offset, animals reported their long vs short judgment at two lateral choice ports. Correct trials yielded a water reward, while incorrect or premature responses produced a white noise sound and a time out. Top view, high-speed video was acquired throughout task performance. (b) Reward contingency. (c) Averaged psychometric curves following bilateral muscimol or saline injections in dorsal striatum (mean ± standard deviation across session means, and logistic fit; n = 3 rats, 4 sessions each). Inset: slope of psychometric curves on consecutive saline and muscimol sessions. All raw data for *Figure 1* can be found in *Figure 1—source data 1*.

The following source data and figure supplement are available for figure 1:

**Source data 1.** The. txt file contains trial by trials stimulus (*Interval*), choice (*choiceLong*), animal (*Name*), treatment (*MuscimolDose*) and session (*Date*).

**Figure supplement 1.** Histological confirmation of cannula placements.

the two curves evolved more slowly so that at the time of interval offset the short preferring subpopulation was still firing at a higher level and a crossing point had not yet been reached (*Figure 2h*).

The observation of large proportions of short- and long-preferring neurons whose dynamics predicted choice is evidence that duration judgments are guided by the state of striatal populations. Might the information afforded by ensembles of striatal neurons account for the pattern of subjects' judgments across all stimuli? To test this hypothesis, we compared session to session fluctuations in behavioral performance with the separability of activity states of simultaneously recorded ensembles at the offset of short as compared to long intervals. Briefly, for each trial in a session we characterized neural population state as a vector $\mathbf{r} = (r_1, r_2,... , r_N)$, where $r_n$ is the number of spikes fired by neuron $n \in [1,N]$ within the last 500 ms of the interval period. Next, for each session we found the linear discriminant that best separated population state vectors according to whether they came from a long or a short interval trial (*Figure 3a*; see Materials and methods). A threshold placed along the linear discriminant was then used as a decision rule (black line in *Figure 3a*) to generate a 'neural duration judgment' for each trial. This procedure allowed us to obtain, for each session, a quantitative description of how well simultaneously recorded neurons could categorize stimuli, i.e., a neurometric function comparable to the behavioral psychometric function (*Figure 3b*). Consistent with duration information being encoded at the population level, we found that for sessions in which greater numbers of neurons were recorded simultaneously (i.e. upper tercile of sessions with regard to population size) psychometric and neurometric performances were similar and strongly correlated ($r^2 = 0.76$, p<0.001; *Figure 3c*). These results demonstrate that a read out of stimulus category from even modestly-sized ensembles of striatal neurons was in many cases sufficient to explain the pattern of duration judgments produced by behaving subjects.

It has been previously reported that duration judgments could be predicted by animals' ongoing behavior during the interval period (*Gouvêa et al., 2014*; *Matthews and Lerer, 1987*; *Killeen and Fetterman, 1988*; *Fetterman et al., 1998*; *Machado, 1997*; *Machado and Keen, 2003*). In addition, it is well known that striatal neurons can fire around movements (*Alexander and Crutcher, 1990*; *Jin and Costa, 2010*). Could the categorization performance of striatal ensembles reflect activity related to movements the animal might be making during the task? To test to what degree ongoing behavior could explain the categorization performance of striatal neural activity, we applied an analogous classification analysis to video images taken of the animal just before interval offset

(see Materials and methods). We found that our ability to categorize intervals using video frames was consistently poorer as compared to neural data collected at the analogous time periods during the task (*Figure 3b*, inset in *Figure 3c*, *Video 1*, *Figure 3—figure supplement 1*, *Figure 3—figure supplement 2a*). In contrast, we were able to categorize stimuli as well as the animal using video frames taken at the point when animals expressed their choice at one of the reward ports (*Figure 3—figure supplement 2b*). Furthermore, movement related responses in the striatum are known to occur both pre- and post-movement onset, much later than in other motor areas such as pre-motor and motor cortex (*Alexander and Crutcher, 1990*). Thus, if purely movement-related activity were responsible for the categorization performance of striatal ensembles, we would expect ensemble performance to display a similar time course to that of video frames. Applying the same analyses at multiple points in time ranging from 500 ms preceding to 500 ms following stimulus offset revealed a strikingly different profile of categorization performance for video frames as compared to neural ensembles (*Figure 3d–e*). Specifically, the time course of duration categorization by neural ensembles was best correlated with the duration categorization by video frames when using spikes collected between 400 ms and 200 ms preceding a reference video frame. These indicate that the categorization performance of striatal neurons was not simply related to the immediate sensorimotor state of the animal, and instead likely reflects that striatal neurons encode an internal neural representation of the state of animals' categorical decisions.

We have shown thus far that categorical information about interval duration contained in the firing of striatal populations at the time of stimulus offset can explain the precision of animals' judgments about duration. However, in the task employed here, categorical judgments must be derived from a continuously evolving decision variable that represents how much time has elapsed since the onset of the stimulus. As indicated by the diversity of firing patterns (*Figure 2d*), the state of population activity evolved continuously during interval stimuli (*Figure 3g*, *Figure 4a*, *Figure 3—figure supplement 3b*), a feature not captured by binary classification. Might trial to trial variations in population state predict the apparent speed of animals' internal representation of elapsed time? To test this possibility, we performed two additional analyses.

First, we projected the state of simultaneously recorded neuronal populations at stimulus offset in individual trials onto the mean trajectory within each session, noted the fraction of the mean trajectory traversed for each trial, and pooled the data for each stimulus over all sessions within a given subject. Indeed, when population state at stimulus offset had advanced relatively more or less along the mean trajectory, animals were more likely to judge intervals as long or short respectively (*Figure 3f–g*, *Figure 3—figure supplement 3*). This effect was observed most consistently for interval stimuli that were closer to the category boundary, and thus variations in projected population state led to horizontal shifts in the psychometric curves (see Materials and methods). These data are consistent with striatal population state encoding a representation of elapsed time that is used by animals to determine categorical judgments. Indeed, such a pattern of population activity has been proposed as a suitable neural code for elapsing time (*Buonomano, 2014*; *Buonomano and Merzenich, 1995*).

However, if such a representation encodes elapsed time, and not only subjects' judgments in this task, neural activity should continuously traverse a non-repeating trajectory in state space in a manner that predicts duration judgments during presentation of particular stimuli. Indeed, even in a low dimensional projection of population activity, we found that network state ran ahead or behind depending on whether the animal judged a near boundary stimulus as long or short (*Figure 4b–c*, *Figure 4—figure supplement 1b,c*, *Figure 4—figure supplement 2b,f,j*). The correspondence between population trajectory and duration judgments further suggests that striatal dynamics may form an internal representation of elapsed time that informed categorical decisions about duration. To directly test this hypothesis, we focused on stimuli near the category boundary and decoded time from the population using a naive Bayes decoder and asked whether such a representation correlated with animals' judgments, exhibiting choice probability (*Britten et al., 1996*). We found that decoded estimates of time ran faster or slower when animals judged a given stimulus as long or short, respectively (*Figure 4d–g*, *Figure 4—figure supplement 1d-g*, cross validated naive Bayes decoder; see Materials and methods). This indicates that striatal activity provides information about elapsing time, the continuously varying decision variable necessary to inform judgments in the task. Furthermore, if this information were read out and used to guide judgments, those judgments would match those of the rats.

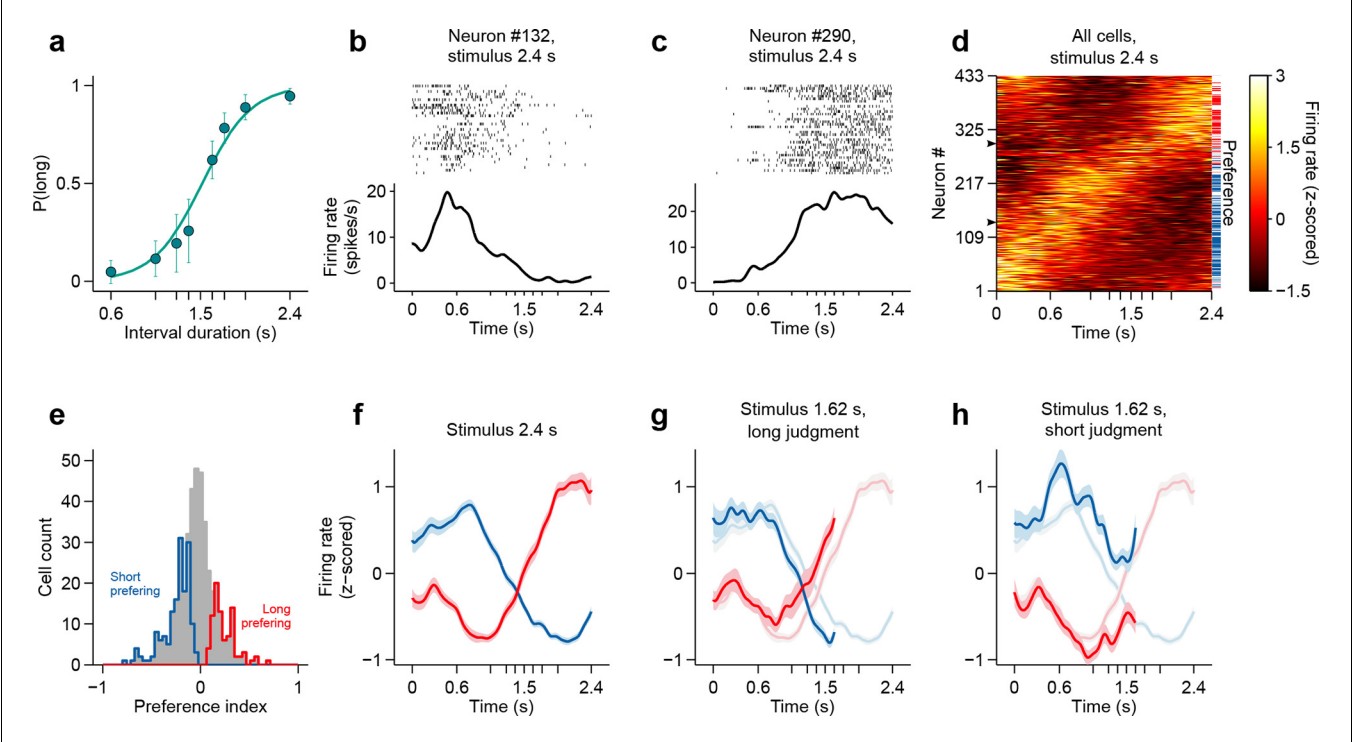

**Figure 2.** Dynamics of striatal subpopulations predict duration judgments. (a) Psychometric function for neural recording sessions (mean ± standard deviation across sessions and logistic fit, n = 37 sessions from 3 rats). (b,c) Raster plot and peri-stimulus time histogram (PSTH) of two example cells for trials in which the longest stimulus interval (2.4 s) was presented. Time = 0 corresponds to stimulus onset. (d) Normalized PSTHs of all neurons for trials in which the longest stimulus interval was presented. Arrowheads indicate cells shown in (b,c). Blue and red ticks indicate cells with significant short and long preferences, respectively. (e) Histogram of preference indices. Blue and red outlines indicate subpopulations with significant short and long preferences, respectively. (f) Averaged, normalized PSTH of the two subpopulations outlined in (e) for trials in which the longest stimulus interval was presented (mean ± SEM). (g) Same as in (f), for trials in which a near-boundary stimulus interval (1.62 s) was judged as long. For comparison, curves shown in (f) are reproduced as a watermark. (h) same as (g) for trials in which the stimulus was judged as short. For single subjects, see *Figure 2—figure supplement 2* . Behavior and neural spike count data for *Figure 2* and *Figure 2—figure supplements 1 and 2* can be found in *Figure 2—source data 1*.

The following source data and figure supplement are available for figure 2:

**Source data 1.** Folder with raw data for *Figures 2–4*.
**Electrophysiological recordings in dorsal striatum.** DOI: 10.7554/eLife.11386.008
**Figure supplement 2.** Dynamics of striatal subpopulations predict duration judgments.

If the striatal activity we describe above directly supported task performance, manipulating the striatum should modify duration judgments. To test whether this was the case, we bilaterally injected the GABAa receptor agonist muscimol (see Materials and methods). As a result, the duration sensitivity of animals' judgments dropped significantly as compared to interleaved saline control sessions (*Figure 1c*; psychometric slope on saline sessions = [1.03 1.20] vs on muscimol sessions = [0.28 0.67]; 95% confidence intervals), yet animals otherwise performed normally. These results, by demonstrating that duration categorization in this task was dependent on a normally functioning striatum, suggest that the neural signals we observed directly supported duration judgments. However, the possibility that muscimol infusions changed other functions important for task performance such as reward processing or memory for the mapping between time and choice can not be ruled out.

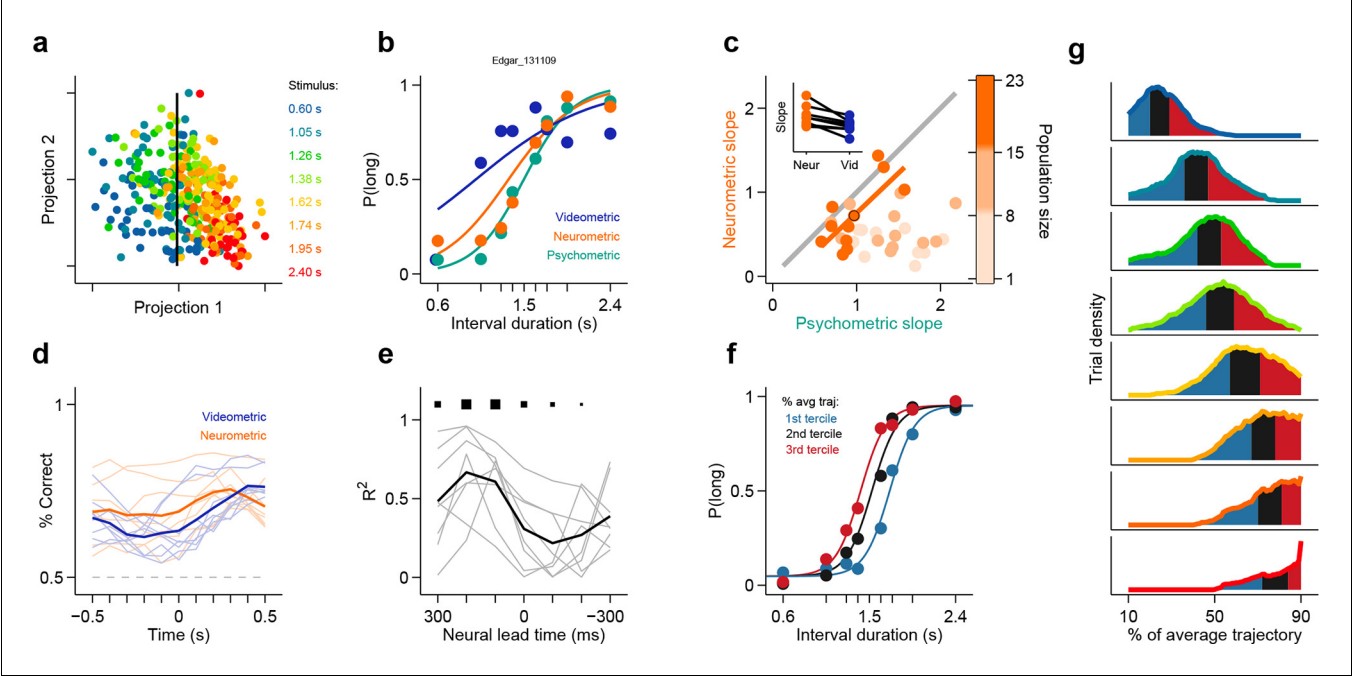

**Figure 3.** Simultaneously recorded population state at interval offset can explain behavioral performance. (a) Low dimensional representation of population state at interval offset for one example session. Black line is the decision rule (see text). (b) Example psychometric, neurometric and videometric curves for the same session as in (a). (c) Slopes of psychometric and neurometric curves for all sessions. Color indicates terciles of population size. Highlighted data point corresponds to the session in (a-b). Inset: regression slope of neurometric and videometric curves for sessions in the upper tercile. See *Figure 3—figure supplement 2* for psychometric-videometric comparison at interval offset and choice. (d) Performance of an ideal observer analysis in predicting stimulus category, applied to neural (orange) and video (blue) data obtained at different times relative to interval offset. Thin lines corresponds to individual sessions. Thick lines are averages. (e) The orange and blue curves (thin lines in panel (d)) for corresponding sessions were regressed against each other at different time shifts. The regression $R^2$ values for each session are shown in thin grey lines. The average over all sessions is shown in black. Sizes of black squares indicate the number of sessions with significant positive correlations (largest squares at 200 and 100 ms correspond to 5 sessions and smallest one at -200 ms to 1, out of a total of 8 sessions). (f) Psychometric curves constructed from trials separated according to whether the population state at stimulus offset had advanced more or less along the mean trajectory. Color indicates terciles shown in (g). (g) Distributions of projection on normalized mean trajectory for all trials for each stimulus are shown (stimuli color coded as in [a]). The equal area bins shown correspond to the groups of trials used for constructing the three psychometric curves shown in panel (f). Data in f-g are from rat Bertrand. See *Figure 3—figure supplement 3* for the remaining two subjects.

The following figure supplements are available for figure 3:

**Figure supplement 1.** Image frames at the end of the neural analysis window do not show a consistent separation between short and long stimulus trials.

**Figure supplement 2.** Behavior at the end of the neural analysis window did not explain the categorization performance of neural populations.

**Figure supplement 3.** Population state at interval offset can explain behavioral performance.

## Discussion

Attempts to understand the neural mechanisms of time estimation have begun to focus on continuously evolving population dynamics as a general mechanism for time encoding across the brain (*MacDonald et al., 2011*; *Mauk and Buonomano, 2004*; *Buonomano, 2014*; *Buonomano and Merzenich, 1995*; *Gershman et al., 2013*). According to this view, time may be encoded by any reproducible pattern of activity across a population of neurons for as long as the pattern is continuously changing and non-repeating. However, no study to date has directly compared the speed of such "population clocks" with the duration judgments of the behaving subjects in which they are found. We show that as rats judged the duration of interval stimuli, striatal neurons displayed dynamics in firing rate that contained information about elapsed time. Furthermore, this information

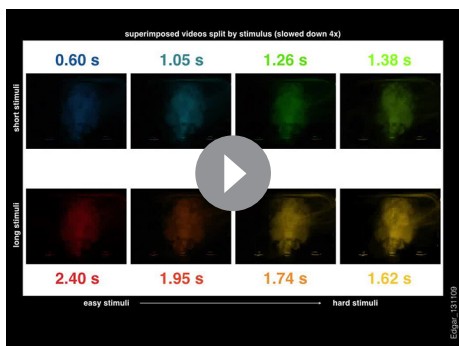

**Video 1.** Video clips from the entire stimulus period do not show a consistent separation between short and long stimulus trials. Superimposed video clips (thresholded and background subtracted) for each stimulus type and for the corresponding stimulus duration (same color conventions as in *Figure 3a* and *Figure 3—figure supplement 1*). Red stop marks signal the end of each video and the corresponding stimulus duration.

was sufficient to account for the animals' perceptual decisions, and was not accompanied by systematic differences in outwardly expressed behavior over time. Combined with the observation that striatal inactivation caused a specific decrement in timing performance, these data suggest that striatal dynamics form a central neural representation of time that guides animals' decisions about duration. Such a coding mechanism in the striatum is well situated to inform the appropriate selection of actions through downstream circuitry involving the globus pallidus, substantia nigra, and various extrinsic connections between the basal ganglia and brainstem, thalamic, and cortical motor areas (*Steiner and Tseng, 2010*). An intriguing question for future studies is how the striatal dynamics we observed during the interval discrimination task are generated. Neurons in multiple cortical layers spread across the entire cortical mantle, as well as thalamic, pallidal, and neuromodulatory populations provide input to the striatum. While time coding has been assessed in some of these populations, a careful analysis of simultaneously recorded populations that might uncover causal relationships between signals in multiple brain areas has not been carried out. Furthermore, local striatal circuitry may also play a role in shaping dynamics. However, the coding properties tested here could be tested in other brain areas where timing signals have been identified such as the hippocampus (*Pastalkova et al., 2008*; *MacDonald et al., 2011*), medial prefrontal (*Kim et al., 2013*; *Xu et al., 2014*), parietal (*Leon and Shadlen, 2003*; *Janssen and Shadlen, 2005*) and motor (*Lebedev et al., 2008*) cortices, and the cerebellum (*Mauk and Buonomano, 2004*), among others. By comparing the signals recorded simultaneously in multiple brain areas during time estimation tasks, it should be possible to identify signatures of functional interaction between brain areas where they exist. Such an approach promises to elucidate where and how time information encoded at the population level is used by the brain to support the myriad time-dependent functions we and other organisms rely on for survival.

## Materials and methods

### Subject

Six male Long-Evans hooded rats (*Rattus norvegicus*) between the ages of 6 and 24 months were used for this study. Three rats were used for neural recordings and three rats for pharmacological manipulations. All experiments were in accordance with the European Union Directive 86/609/EEC and approved by the Portuguese Veterinary General Board (Direcção-Geral de Veterinária, project approval 014303 - 0420/000/000/2011).

### Behavior

Rats were trained to perform a previously described two-alternative forced-choice timing task (*Gouvêa et al., 2014*). Briefly, animals had to categorize time intervals as either long or short by making left/right choices. For each session the animals were placed in a custom made behavioral box containing 3 nose ports and a speaker. Each trial was self-initiated by entry into the central nose port and was followed by a pair of brief auditory tones (square pulses at 7,500 Hz, 150 ms) separated by an interval selected randomly out of 8 possible durations (0.6, 1.05, 1.26, 1.38, 1.62, 1.74, 1.95 and 2.4 s). Judgments were reported at two laterally located nose ports. Left responses were reinforced with a drop of water (solenoid valves, Lee Company) after intervals longer than 1.5 seconds, and right responses otherwise. Incorrect responses were punished with a brief white noise

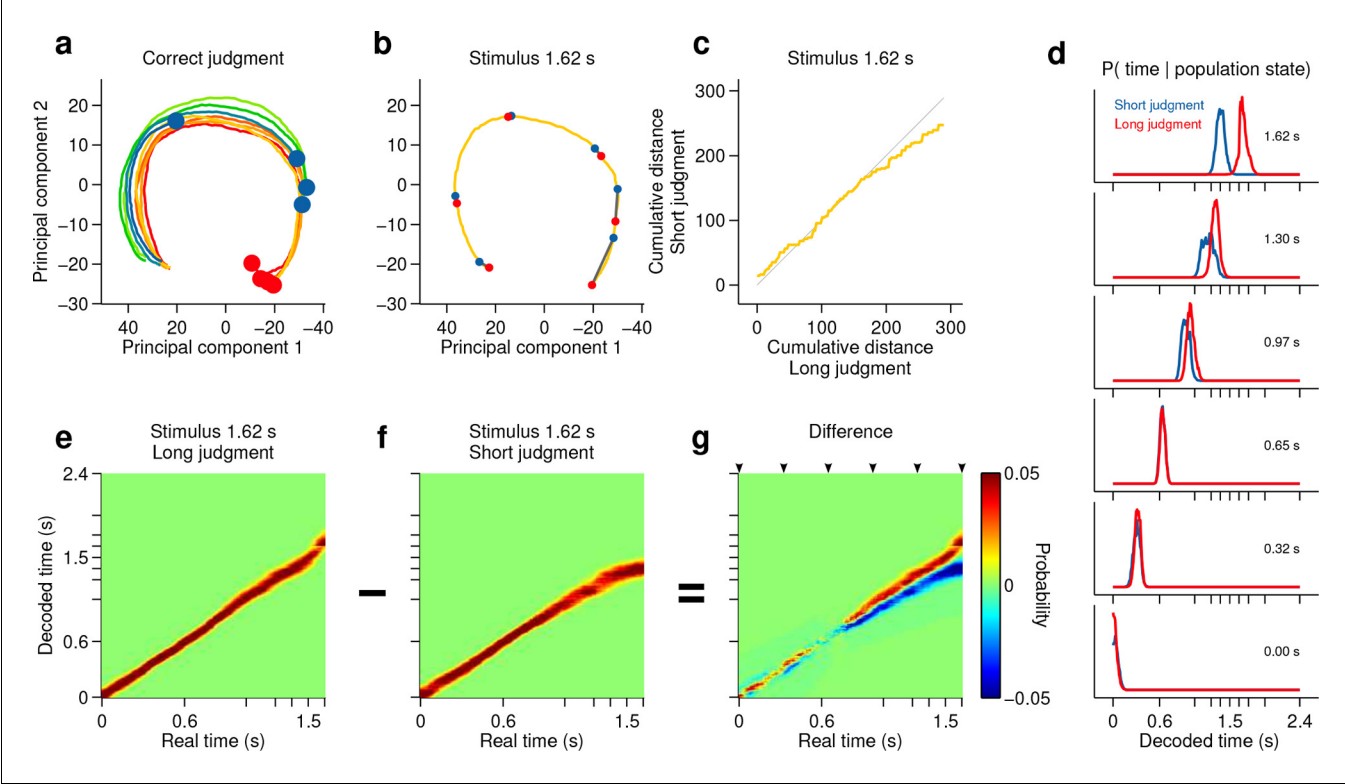

**Figure 4.** Smoothly changing population state encodes elapsing time in accordance with perceptual report for a long stimulus. (**a**) Low dimensional representation of population state during entire interval period of correct trials. Line colors indicate interval duration (warmer colors are longer intervals, as in *Figure 3a*). Dots are placed at the interval offset end, and their color indicates judgment (blue: short; red: long). (**b–g**) Population state and decoded time for a single long, near boundary stimulus interval (1.62 s). (**b**) Yellow curve is same as in (**a**). Red dots are 6 time points evenly spaced between interval onset and offset. Blue dots are projections of population state during short judgment trials. Grey lines link population states at equivalent time points. (**c**) Average cumulative distance travelled in full neural space along trajectory represented in (**b**) on long versus short judgment trials. (**d**) Posterior probability of time given population state at the time points indicated in (**b**), averaged within trials of each judgment type. (**e,f**) Same as (**d**) for the entire interval period. (**g**) Difference between posteriors for long and short judgment trials. Arrowheads indicate same time points used in (**b,d**). n = 433 neurons from 3 rats. See *Figure 4—figure supplement 1* for a different near boundary stimulus, and *Figure 4—figure supplement 2* for data from individual subjects.

The following figure supplements are available for figure 4:

**Figure supplement 1.** Smoothly changing population state encodes elapsing time in accordance with perceptual report for a short stimulus.

**Figure supplement 2.** Single subjects show smoothly changing population states that encode elapsing time in accordance with perceptual report.

burst (150 ms) and a time out (10 s). High speed video (120 fps) was collected from above during task performance. Psychometric functions were fitted using the following two-parameter logistic function

$$f(x) = \frac{1}{1 + e^{-b(x-c)}}$$

where *b* controls the slope and *c* is the inflection point of the curve.

## Electrophysiology

Rats were implanted with 32-channel tungsten microwire moveable array bundles (*Figure 2—figure supplement 1a*, Innovative Neurophysiology) under isoflurane anaesthesia. All recordings (*Figure 2—figure supplement 1*) targeted dorsal striatum with coordinates centred at +0.2 mm AP and ± 3 mm ML (rat Bertrand), and +0.84 mm AP and ± 2.5 mm ML (rats Edgar and Fernando), from Bregma. Rats were given a week of post-surgical recovery and array placements were confirmed

with histology (*Figure 2—figure supplement 1c*). Neural signals were recorded at 30 kHz during behavior, amplified and band-pass filtered at 250–750 Hz (Cerebus - Blackrock Microsystems). Each independent bundle was moved 50-100 µm after every recording session to ensure that independent neural populations were sampled across recording sessions. Waveforms corresponding to action potentials from single neurons were sorted offline using principal component analysis (PCA) (offline sorter, Plexon). All remaining analysis were run in Matlab (Mathworks) software. We selected all isolated units with a mean session firing rate >0.5 Hz and from sessions with >70% correct performance (averaged across all stimuli) and a minimum of 250 trials (n=433 cells, 37 recording sessions, 3 animals; rat Bertrand: 136 units, 10 sessions; rat Edgar: 163 units, 9 sessions; rat Fernando: 134 units, 18 sessions). The general result was found in all subjects. Sample size was not computed during study design. To build PSTHs, spikes were counted in 2-ms bins and convolved with a Gaussian kernel with 25-bin standard deviation. PSTHs in *Figure 2d* were ordered by angular position in the space formed by the first 2 principal components describing firing dynamics (i.e., dimensions are all time bins within interval period, samples are each neuron's mean PSTH). This method (*Geffen et al., 2009*) orders cells with respect to their dynamics while taking into consideration the full response profile over the relevant temporal window, and not just a single response feature such as peak response time.

## Pharmacology

We implanted 3-mm 20-gauge stainless steel guide cannulae (Bilaney) bilaterally into the striatum of 3 rats [+0.84 mm anterior-posterior (AP), ± 2.5 mm medial-lateral (ML), from Bregma, and -3mm dorsal-ventral (DV, from cortex surface) under isoflurane anesthesia. After one week of post-surgical recovery and 4 days of training, rats were injected with either vehicle (saline, PBS 1x) or muscimol (GABA-A agonist, 100 mg/L (rats Albert and Yuri) and 300 mg/L (rat Zack), Sigma™) solutions in four alternate days. Two 1-µL syringes (Hamilton), attached to an injection pump (Harvard Apparatus) through 20-gauge internal cannulae that extended 1.5 mm bellow the guide cannulae, injected 0.6 µL of solution per site during 2.5 min. The internal cannulae were left in place for an additional 1.5 min and the rats were given a 45-min recovery period in their home-cage before starting the task. Cannula placements were confirmed by histology (*Figure 1—figure supplement 1*). The general result was found in all sessions of all subjects. Sample size was not computed during study design.

## Preference index

We counted spikes during the last 500 ms of the stimulus period, and built two separate spike count distributions for short and long judgment trials. Next, we used a ROC analysis to measure the separation between distributions (95% bootstrap confidence interval, 1000 iterations). We then transformed the area under the ROC curve ($auROC \in [0,1]$) into a preference index ($k = 2*auROC - 1$; $k \in [-1,1]$). We adopted the convention that neurons with positive preference indices fired preferentially for long stimuli (*Figure 2e*).

## Low dimensional representations of population state

We refer to the vector describing instantaneous firing rates (measured within 500-ms wide, 10-ms apart, overlapping time bins) across a population of neurons as the population state. The population state vector is a high dimensional variable (i.e., it has as many dimensions as neurons). With the purpose of visualizing population state in 2d plots, we employed standard dimensionality reduction techniques. In *Figure 3a*, we chose to represent in the abscissa a direction that emphasizes the separability between short and long stimulus trials (i.e., the direction that maximizes variance between groups while minimizing variance within groups; Fisher's linear discriminant; see below), and in the ordinate the axis of maximal variance that is also orthogonal to the abscissa (i.e. first principal component calculated in the null space of the linear discriminant). In *Figure 4a–b*, population state is represented in the space formed by the first 2 principal components describing population state, calculated during presentation of the interval for which choice variance is maximal (i.e. dimensions are neurons, samples are averaged spike counts for the time bins within that interval).

## Neurometric curves

For each trial in a session we characterized neural population state as a vector $\mathbf{r}$ = ($r_1$, $r_2$,... , $r_N$), where $r_n$ is the number of spikes fired by neuron $n \in [1,N]$ within the last 500 ms of the interval period in that trial. Next, for all trials but one from each session (*training set*; leave-one-out cross-validation procedure), we found the linear discriminant that best separated population state vectors according to whether they came from long or short interval trials (Fisher's linear discriminant analysis, LDA). The linear discriminant is given by

$$\mathbf{w} = argmax \frac{\mathbf{w}^T \mathbf{S}_B \mathbf{w}}{\mathbf{w}^T \mathbf{S}_W \mathbf{w}} = S_W^{-1}(\mu_1 - \mu_2)$$

where $\mathbf{w}$ is the vector of coefficients for the linear discriminant, $S_B$ is the between class covariance, $S_W$ the within class covariance and $\mu_1$ and $\mu_2$ are the means of all points in class 1 and class 2 respectively. A threshold placed along the linear discriminant was then used as a decision rule applied to neural data from the remaining trial (*test set*). *Figure 3a* depicts population vectors from an example session (projection 1: linear discriminant, no cross-validation; projection 2: first principal component of the orthogonal subspace; black line: decision rule). We iterated over this procedure until all trials had been tested, thus obtaining for each trial a 'neural duration judgment'. In analogy with behavioral judgments, we used two parameter logistic fits to obtain a quantitative description of the performance of simultaneously recorded neurons in categorizing stimuli -the neurometric function (*Figure 3b*, orange curve).

## Videometric curves

Full session videos (256x192 pixels resolution) were cut into 3-s long clips with Bonsai (*Lopes et al., 2015*). Individual frames from approximately 75 ms before interval offset were used for this analysis (*Figure 3—figure supplement 1*). This buffer was added to ensure that all frames used preceded stimulus offset. Images were first represented as vectors composed of individual pixel luminance values. Given that image sequences tend to lie on curved low dimensional manifolds in pixel space (*Pless, 2003*), any slight differences in behavioral state reflected in images collected at the offset of short and long interval categories are not necessarily expected to be linearly separable. Thus, we employed isomap (*Tenenbaum and Silva, 2000*), a non-linear dimensionality reduction method, to obtain an information rich yet low dimensional representation of animals' ongoing behavior. This approach has the advantage over tracking methods that it does not make assumptions as to what part of the animals' movements might provide information about stimulus category. The neighborhood size, used to compute the shortest paths between data points, was set to 25 frames to minimize, on average, the dimensionality at which the reconstruction error elbow occurred. In analogy with the neurometric curves, for each stimulus type, we then trained a linear discriminant (leave-one-out cross-validation procedure) to classify frames into those that were recorded during trials where a 'short' or 'long' stimulus interval was presented. The classification was performed in the reduced space determined by isomap. As a positive control for the method, we repeated the same analysis for frames captured at the moment animals expressed their judgment by inserting their snout at one of the two choice ports (*Figure 3—figure supplement 2*). Here, the neighborhood size was chosen to be the minimum for which all frames (from a single session) could be included in a single embedding. This analysis was done for all usable videos (8 out of 11) of sessions in the upper tercile with regard to population size.

## Time course of classification performance from neural and video data

To compare how the decoding performance using neural and video data evolved over time, the classification analyses described in Neurometric curves and Videometric curves was performed every 100 ms within a one second window centered around stimulus offset. Video frames at the each time point and neural data in a 200 ms time bin terminating at each time point were used for the analysis. This generated neural and video classification curves that described the ability of simultaneously recorded neural ensembles and video frames to correctly classify interval stimuli as long or short (*Figure 3d*). To determine the relative timing of classification ability in neural ensembles and behavior, we regressed the neural classification curve against the video classification curve for shifts ranging from −300 ms to 300 ms in 100 ms steps (*Figure 3e*).

## Psychometric curves split by population state at interval offset

We projected neural activity (of populations composed of simultaneously recorded neurons) on individual trials in high dimensional neural space onto the mean trajectory of those neurons during the delay period for correct trials. Neural activity was defined as the vector of firing rates of the population obtained by convolving spike trains using a causal kernel given by gamma density function with parameters $\theta$ = 100 ms and $k$ = 2. We normalized these projections by the length of the mean trajectory of that group of neurons for the longest interval. Pooling normalized projections over all sessions for each animal, we plotted, for each stimulus, distributions of normalized projections at interval offset. To test whether distance traversed along the mean trajectory is predictive of animals' perceptual report, we separated the distribution of pooled projections for each stimulus into 3 bins. Psychometric curves were constructed using trials from each bin. To quantify the key differences between each of these psychometric curves, we performed model comparison using the following 4 parameter logistic function

$$f(x) = d + \frac{a - d}{1 + e^{-b(x-c)}}$$

where $b$ controls the slope, $c$ is the inflection point and $a$ and $d$ are the maximum and minimum values of the curve respectively. For two of three animals (Bertrand and Edgar), the model that best accounted for the differences between the three curves (based on Bayesian Information Criterion (BIC) scores) was one with only horizontal shifts between the curves. In the third animal (Fernando), the model that best fit the data was one in which the fit to the three curves differed in both horizontal shift and slope.

A trial's projection on the mean trajectory can be interpreted as a method for decoding time from neural state. Hence, trials that are outliers in the distribution of projections on the mean could potentially correspond to trials where the animal was disengaged. To remove such trials we defined a fraction (60%) of normalized trajectory around the mode of the distribution of pooled projections for each stimulus and excluded trials with projections outside this window.

## Population decoder

We decoded elapsed time from striatal population activity using a cross validated, flat prior naive Bayes decoder. For each neuron $n \in [1, N]$, spike counts $r_n$ were observed in 500-ms wide, 10-ms apart, overlapping time bins within the interval period (time referring to the right edge of the bin). For a given $r_n$, the probability that the current time is $t$ was estimated as the likelihood of observing $r_n$ spikes at time $t$:

$$P(t|r_n) \propto P(r_n|t)$$

To obtain the likelihood term $P(r_n \mid t)$, we estimated the joint distribution $P(r_n, t)$ by computing, for each time bin, a weighted histogram of spike counts across all correct trials. For trials in which stimulus interval $i$ was presented, spike counts contributed to the histogram with weight $w_i$ defined as the normalized choice variance associated with that interval,

$$w_i = P(C_S|i)P(C_L|i)$$

where $C_S$ and $C_L$ indicate short and long choices respectively. As a result, near boundary interval trials had a greater influence on the estimate of the joint distribution. Histograms were then smoothed using local linear regression (lowess) and normalized to unit area. When decoding from correct trials, leave-one-out cross validation was implemented by computing the joint distribution from all correct trials but the decoded one; incorrect trials were decoded using an estimate of the joint distribution computed from all correct trials. Multi-session population state vectors $\mathbf{r} = (r_1, r_2, \ldots, r_N)$ were obtained by concatenating together data from trials of same stimulus and choice type. By assuming statistical independence between spike counts of different neurons in $\mathbf{r}$, we could compute population estimates of $t$ as the product of single neuron estimates:

$$P(t|\mathbf{r}) \propto \prod_{n=1}^{N} P(r_n|t)$$

Data presented is the average over 100 random concatenations.

## Acknowledgements

We would like to thank Bassam Atallah, Brian Lau and Masayoshi Murakami for reading earlier versions of the manuscript, Gustavo Mello for help with surgeries, the Champalimaud Research Vivarium and Histology platforms, Serkan Sülün for pre-processing of video data, João Frazão and Gonçalo Lopes for help with *Video 1*, João Semedo for technical discussions regarding Isomap, and Bruno Ceña for logistic support.

## Additional information

### Funding

| Funder | Grant reference number | Author |
|---|---|---|
| Champalimaud Foundation | | Thiago S Gouvêa<br>Tiago Monteiro<br>Asma Motiwala<br>Sofia Soares<br>Christian Machens<br>Joseph J Paton |
| Simons Foundation | SCGB #325476 | Tiago Monteiro<br>Joseph J Paton |
| Bial Foundation | BIAL-BIC-188/12 | Thiago S Gouvêa<br>Tiago Monteiro<br>Sofia Soares<br>Joseph J Paton |
| Portuguese Foundation for Science and Technology | | Thiago S Gouvêa<br>Asma Motiwala<br>Sofia Soares |

The funders had no role in study design, data collection and interpretation, or the decision to submit the work for publication.

### Author contributions

TSG, TM, JJP, Conception and design, Acquisition of data, Analysis and interpretation of data, Drafting or revising the article; AM, SS, CM, Analysis and interpretation of data, Drafting or revising the article

### Author ORCIDs

Thiago S Gouvêa, http://orcid.org/0000-0002-0727-5838
Tiago Monteiro, http://orcid.org/0000-0002-2836-8961
Asma Motiwala, http://orcid.org/0000-0002-7693-2731
Sofia Soares, http://orcid.org/0000-0002-4594-0202
Christian Machens, http://orcid.org/0000-0003-1717-1562
Joseph J Paton, http://orcid.org/0000-0002-7693-2731

### Ethics

Animal experimentation: All experiments were in accordance with the European Union Directive 86/609/EEC and approved by the Portuguese Veterinary General Board (Direcção-Geral de Veterinária, project approval 014303 - 0420/000/000/2011)

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
