## [Decision Letter]

Thank you for submitting your work entitled "Striatal dynamics explain duration judgments" for consideration by *eLife*. Your article has been reviewed by three peer reviewers, and the evaluation has been overseen by Timothy Behrens as the Senior and Reviewing Editor.

The following individual involved in review of your submission has agreed to reveal his identity: Patrick Simen (Reviewer).

The reviewers have discussed the reviews with one another and the Reviewing editor has drafted this decision to help you prepare a revised submission.

Summary:

This paper addresses an important and understudied problem: how does the brain encode time on the scale of a second or so? The authors use a behavioral task in which rats are trained to nose poke left or right depending on the interval between two auditory tones – specifically, whether the interval was short or long in relation to a 1.5 s reference. Striatal recordings from hundreds of cells revealed a robust temporal code: it was possible to determine elapsed time from the onset of the trial based on the current population vector with a high degree of accuracy. Furthermore, it was possible to predict the incorrect behavioral responses based on whether the population trajectory was "running early" or "running late". Together with timing-effects of muscimol infusions in striatum (although see concerns below), these data provide and important physiological link between striatal processing and sensory timing.

Essential revisions:

1) The reviewers raised serious concerns about the muscimol data.

First, the data are from two rats. The reviewers do not think it is easy to make interpretations from two animals and ask that you increase the number. Whilst it is *eLife* policy not to ask for major new data acquisition under revision, the reviewing editor agrees that this is important to solidify the results in this case and believe that you could collect the data within the two month framework. The reviewers think that a good benchmark for a muscimol experiment is 5-6 animals. Please let us know early if you will not be able to perform the extra data acquisition.

Second, the reviewers were concerned that it is very difficult to tell where the cannulae are from the graphic (they are many hundreds of microns wide) – they might very well be in the dorsolateral striatum. Actual histology would be better.

Lastly in this section, there were questions about the behavioural effects of muscimol.

Why didn't muscimol systematically retard or advance the timer if it is truly a clock? It appears that muscimol infusions just increased the variability of the choices, but that doesn't mean that the striatum isn't just listening to a "clock" somewhere else in the brain. I would love to see that a striatal clock could, for example be physiologically sped up or slowed down, as in the amphetamine studies of Meck and colleagues, but I don't think that's what's shown here. Given the role of striatum in reward-guided behavior, I could imagine that inactivating it just makes the task not worth paying attention to, or that the decision stage is disrupted while the "clock", wherever it is, is still intact.

2) With respect to the video analysis and controls for movement.

The reviewers thought that the video analysis likely did a good job controlling for motion effects, but did not think they were clearly presented and had some specific concerns (see point from reviewer directly below). In the discussion, one reviewer suggested including a video that could make the classification clearer. There were also suggestions that the claim could be strengthened by including basic motor control tasks (open-field, circling, paw-rearing) in the muscimol animals, but in the discussion it was agreed that whilst such data would be convincing if you have it, this is not essential.

Specific point from reviewer:

I don't understand the author's valiant effort to use classifiers to analyze movement. Traditionally, force-controlled or highly stereotyped motor tasks (see work by Whishaw) are required – in this domain, video assessing kinematics is useful. As presented, I'm not sure if we're looking at movement preparation, licking, reward expectation, etc. See relevant work by Cowan et al., and David Redish on the issue of 'embodied' movement. The animals' behavioral reports are most helpful; as these at least have some consistent events.

3) There were also questions pertaining to the analysis and interpretation of the neural data.

I don't think the evidence is particularly strong that time is specifically encoded in the striatum in the way proposed. It seems plausible to me that the striatal neurons are just encoding an output stage of the decision process, in which a short or long choice is preferred. Yes, it is possible to decode subjective time from it, but I would imagine you could do that with population activity recorded from many different brain areas (in particular, areas examined in several of the papers cited here).

and relatedly:

This paper provides nice support for the notion that time is encoded in the evolving population dynamics of neurons. The authors also provide some evidence that the striatum is directly involved in the task. Less clear, and admittedly far outside the scope of this paper or any other single paper, is whether the code/dynamics is generated within the striatum. A few sentences as to the authors' thoughts on this issue would be useful, and help guide future research.

With respect to the following comment from one reviewer – in the discussion the reviewers and editor agreed that the figures could be clarified, but leave it up to you whether you think this can be best achieved using a PCA.

The neuronal analyses are unclear to me. In Figure 2, The rasters are helpful, as are the averages, but I'd favor something like PCA to help explain the dataset, as it seems like all patterns are represented in their large population (Figure 2). For Figure 3 is the color population size in terms of neurons? I think I don’t understand what is being plotted in Figure 3? I'd like more detail on the 'neurometric' ideal observer – it'd be nice to see the predictions for a single session, and for single population, and even a single neuron as I'm not sure how much neuron population size matters (is that Figure 3?), and I'm not sure what the 'density' in Figure 3 shows – or the corresponding plots in the supplement. At the end of the day, is it surprising that striatal neurons predict temporal durations? Also, could the authors define 'psychometric'? It's not clear what this means. Figure 3 likely needs to be greatly clarified for most audiences.

4) The reviewers were not clear on where in the striatum the authors are interested in. The author's electrodes span the whole striatum, and dorsolateral striatum is completely different than dorsomedial striatum. See work from Henry Yin. Some electrodes even seem to reach ventral striatum.

---

## [Author Response]

Essential revisions:

1) The reviewers raised serious concerns about the muscimol data.First, the data are from two rats. The reviewers do not think it is easy to make interpretations from two animals and ask that you increase the number. Whilst it is eLife policy not to ask for major new data acquisition under revision, the reviewing editor agrees that this is important to solidify the results in this case and believe that you could collect the data within the two month framework. The reviewers think that a good benchmark for a muscimol experiment is 5-6 animals. Please let us know early if you will not be able to perform the extra data acquisition.

We have now included additional experiments involving the infusion of muscimol into the striatum during task performance. This brings the total numbers to 6 muscimol and 6 saline sessions performed on adjacent days in three different animals (2 saline and 2 muscimol infusions in each animal). In all individual sessions, muscimol infusions produced a significant decrease in the sensitivity of animals’ judgments to interval duration as compared to saline infusions performed on the previous day. It is important to point out that these are reversible inactivations, not permanent lesions, allowing us to collect multiple control and test conditions within each animal. Thus, we believe there should be no doubt that these results reflect a general effect of infusing muscimol on task performance. However, given the slow time course of pharmacology, interpretive issues do remain (see below).

Second, the reviewers were concerned that it is very difficult to tell where the cannulae are from the graphic (they are many hundreds of microns wide) – they might very well be in the dorsolateral striatum. Actual histology would be better.

We have now included histology and have updated our schematics in Figure 1—figure supplement 1 to reflect the width of the infusion cannulae above the striatum.

Lastly in this section, there were questions about the behavioural effects of muscimol.

Why didn't muscimol systematically retard or advance the timer if it is truly a clock? It appears that muscimol infusions just increased the variability of the choices, but that doesn't mean that the striatum isn't just listening to a "clock" somewhere else in the brain. I would love to see that a striatal clock could, for example be physiologically sped up or slowed down, as in the amphetamine studies of Meck and colleagues, but I don't think that's what's shown here. Given the role of striatum in reward-guided behavior, I could imagine that inactivating it just makes the task not worth paying attention to, or that the decision stage is disrupted while the "clock", wherever it is, is still intact.

The finding that striatal muscimol infusions reduced the sensitivity of animal’s judgments only shows that signals contributed by the striatum are necessary for normal task performance. Muscimol could have had an effect by changing reward processing, memory for a mapping between time and choice, and/or by making less reliable the representation of the decision variable, elapsed time, that is the major focus of the physiology in the paper. We have now included language explaining this interpretive issue (Results). The experiment that the reviewer seems to be asking for is one wherein we systematically speed or slow the spatiotemporal patterns of activity within striatal populations. Indeed we would love to do such an experiment, although accomplishing this is not trivial and furthermore we believe it is beyond the scope of this paper. Such an experiment would require transiently altering spatiotemporal population dynamics on a small fraction of trials so as to prevent the animal from adapting its choices to the reward feedback it might receive from altered patterns of judgment.

2) With respect to the video analysis and controls for movement.

The reviewers thought that the video analysis likely did a good job controlling for motion effects, but did not think they were clearly presented and had some specific concerns (see point from reviewer directly below). In the discussion, one reviewer suggested including a video that could make the classification clearer.

We have now included a supplementary figure and Video 1 to give readers a sense of ongoing behavior during the task.

There were also suggestions that the claim could be strengthened by including basic motor control tasks (open-field, circling, paw-rearing) in the muscimol animals, but in the discussion it was agreed that whilst such data would be convincing if you have it, this is not essential.

We did not collect such data, although we agree that it might be informative to collect in the future.

Specific point from reviewer:

I don't understand the author's valiant effort to use classifiers to analyze movement. Traditionally, force-controlled or highly stereotyped motor tasks (see work by Whishaw) are required – in this domain, video assessing kinematics is useful. As presented, I'm not sure if we're looking at movement preparation, licking, reward expectation, etc. See relevant work by Cowan et al., and David Redish on the issue of 'embodied' movement.

We are well aware of the possibility that embodied strategies could play an important role in many instances of timing as well as other cognitive processes. In fact we previously published a paper that examined snout position in relation to timing judgments. However, we were underwhelmed by the power of tracking approaches to detect more subtle differences in behavior that might explain variance in neural firing collected during our experiments. By treating individual images as points in the space defined by the luminance of all of the pixels in the frame, we avoid a pitfall of selective annotation of videos, computer vision, etc. which only consider a subset of the information contained in the image. This was our motivation for the approach to video analysis contained in the paper. We point out that animals cannot lick at the reward ports during the task epochs that we analyze, as this would result in an aborted trial.

The animals' behavioral reports are most helpful; as these at least have some consistent events.

We agree with the reviewer that the behavioral reports are helpful, indeed they play a central role in most of the analyses of neural data in relation to behavior currently contained in the paper.

3) There were also questions pertaining to the analysis and interpretation of the neural data.

I don't think the evidence is particularly strong that time is specifically encoded in the striatum in the way proposed. It seems plausible to me that the striatal neurons are just encoding an output stage of the decision process, in which a short or long choice is preferred.

Were striatal neurons just encoding the output of a decision, we would expect neural activity to occupy two states, with perhaps a transition period in between those two states. We do not observe such a pattern, as evidenced by the distribution of response profiles over time (Figure 2, Figure 2—figure supplement 2) and our ability to decode time continuously (Figure 4, Figure 4—figure supplement 1 and Figure 4—figure supplement 2) from the population. That said, we do not claim that information about the animals’ decisions is not also contained in the population.

Yes, it is possible to decode subjective time from it, but I would imagine you could do that with population activity recorded from many different brain areas (in particular, areas examined in several of the papers cited here).

This sentence seems to directly conflict the previous one, wherein the reviewer proposes that the striatum only encodes the output of a decision process. That aside, we do not doubt that signals related to elapsed time could be recorded from other brain areas. Time is fundamental to many diverse functions that the brain is tasked with carrying out.

However, a representation of time is necessary for the assumed functionality of the basal ganglia in learning to associate stimuli and actions with reward. Our work is the first to demonstrate that information about elapsed time contained in the evolving population dynamics of striatal neurons reflects time information used by the animals’ to guide their duration judgments.

and relatedly:

This paper provides nice support for the notion that time is encoded in the evolving population dynamics of neurons. The authors also provide some evidence that the striatum is directly involved in the task. Less clear, and admittedly far outside the scope of this paper or any other single paper, is whether the code/dynamics is generated within the striatum. A few sentences as to the authors' thoughts on this issue would be useful, and help guide future research.

We agree with the reviewer that this issue could be discussed in greater depth and we have now added text in the Discussion to expand on this important point.

With respect to the following comment from one reviewer – in the discussion the reviewers and editor agreed that the figures could be clarified, but leave it up to you whether you think this can be best achieved using a PCA.

The neuronal analyses are unclear to me. In Figure 2, The rasters are helpful, as are the averages, but I'd favor something like PCA to help explain the dataset, as it seems like all patterns are represented in their large population (Figure 2).

Indeed, Figure 4 and Figure 4—figure supplement 2 contain projections of the neural data onto the first two principal components of the data.

For Figure 3 is the color population size in terms of neurons?

Yes, population size refers to the size of the simultaneously recorded population on each session.

*I think I don’t understand what is being plotted in Figure 3? I'd like more detail on the 'neurometric' ideal observer –*

We direct the reviewer’s attention to the section of the Materials and methods titled ‘neurometric curves’ for a detailed description of the ideal observer analysis.

it'd be nice to see the predictions for a single session, and for single population,

We agree that it is nice to see single session examples, which in this case are also single population examples since only simultaneously recorded populations are included in this analysis. Indeed we had included such a single session, single population, example in Figure 3.

and even a single neuron as I'm not sure how much neuron population size matters (is that Figure 3?),

The single neuron ideal observer analysis corresponds to the receiver operating characteristic analysis displayed in Figure 2 and extended Figure 2—figure supplement 2.

and I'm not sure what the 'density' in Figure 3 shows – or the corresponding plots in the supplement.

We have now updated the figure to label the axis in question as “Trial density” to make it more clear that density refers to the density of single trial population states as a function of the fraction of mean neural trajectory traversed.

At the end of the day, is it surprising that striatal neurons predict temporal durations?

We believe it is less surprising that striatal, or any other, neurons predict time than that they predict time in a manner that agrees with the trial to trial variations in temporal judgments exhibited by a behaving subject. Furthermore, many models for how time is encoded by neural systems exist, and our data provides support for a subset of these models. Hence, if the reviewer is asking whether our data is surprising in the sense that it provides information about how timing on the scale of seconds is accomplished by the brain, then yes, we believe these results are surprising and thus informative.

Also, could the authors define 'psychometric'? It's not clear what this means.

The psychometric function, a standard term in sensory-based decision tasks, is an “inferential model applied in detection and discrimination tasks. It models the relationship between a given feature of the physical stimulus, e.g. velocity, duration, brightness, weight etc., and the forced-choice responses of the subject.”

Figure 3 likely needs to be greatly clarified for most audiences.

Without more specific instructions, we are not sure how to proceed beyond the changes we list above.

4) The reviewers were not clear on where in the striatum the authors are interested in. The author's electrodes span the whole striatum, and dorsolateral striatum is completely different than dorsomedial striatum. See work from Henry Yin. Some electrodes even seem to reach ventral striatum.

We have now included text highlighting our interest in the dorso-central striatum, an area where lesions have been shown previously to produce timing deficits (Results, second paragraph).